# Karyological Diversification in the Genus *Lyciasalamandra* (Urodela: Salamandridae)

**DOI:** 10.3390/ani11061709

**Published:** 2021-06-08

**Authors:** Marcello Mezzasalma, Gaetano Odierna, Agnese Petraccioli, Michael Veith, Fabio Maria Guarino

**Affiliations:** 1Department of Biology, University of Naples Federico II, I-80126 Naples, Italy; m.mezzasalma@gmail.com (M.M.); petra.ag@gmail.com (A.P.); fabio.guarino@unina.it (F.M.G.); 2CIBIO-InBIO, Centro de Investigação em Biodiversidade e Recursos Genéticos, InBIO, Universidade do Porto, Rua Padre Armando Quintas 7, 4485-661 Vairaõ, Portugal; 3Department of Biogeography, Trier University, Universitätsring 15, 54296 Trier, Germany; veith@uni-trier.de

**Keywords:** amphibia, chromosome banding, evolution, heterochromatin, karyotype, NOR heteromorphism, phylogenetic diversification

## Abstract

**Simple Summary:**

The Lycian salamanders of the genus *Lyciasalamandra* are characterized by a debated taxonomy and phylogenetic relationships. They have been the subject of various molecular and phylogenetic analyses, but their chromosomal diversity is completely unknown. We here present a comparative cytogenetic analysis on five out of the seven described species and seven subspecies of *Lyciasalamandra*, providing the first karyological assessment on the genus and comparing them to closely related representatives of the genus *Salamandra*. We analyzed the occurrence and distribution of different conserved (chromosome number and morphology) and highly variable karyological features. We found an impressive diversity in the configuration of nucleolus organizing regions (NORs), which alternatively occur either as heteromorphic or homomorphic loci on distinct regions of different chromosome pairs. We highlight that the observed peculiar taxon-specific pattern of chromosome markers supports the taxonomic validity of the different studied evolutionary lineages and is consistent with a scenario of synchronous evolution in the Lycian salamanders.

**Abstract:**

We performed the first cytogenetic analysis on five out of the seven species of the genus *Lyciasalamandra*, including seven subspecies, and representatives of its sister genus *Salamandra.* All the studied species have a similar karyotype of 2n = 24, mostly composed of biarmed elements. C-bands were observed on all chromosomes, at centromeric, telomeric and interstitial position. We found a peculiar taxon-specific NOR configuration, including either heteromorphic and homomorphic NORs on distinct regions of different chromosomes. *Lyciasalamandra a.*
*antalyana* and *L*. *helverseni* showed two homomorphic NORs (pairs 8 and 2, respectively), while heteromorphic NORs were found in *L. billae* (pairs 6, 12), *L. flavimembris* (pairs 2, 12), *L. l. luschani* (pairs 2, 12), *L. l. basoglui* (pairs 6, 12), *L. l. finikensis* (pairs 2, 6) and *S. lanzai* (pairs 8, 10). Homomorphic NORs with an additional supernumerary site were shown by *S*. *s*. *salamandra* (pairs 2, 8) and *S. s*. *gigliolii* (pairs 2, 10). This unexpected highly variable NOR configuration is probably derived from multiple independent NOR translocations and paracentric inversions and correlated to lineage divergence in *Lyciasalamandra.* These results support the taxonomic validity of the studied taxa and are consistent with a hypothesized scenario of synchronous evolution in the genus.

## 1. Introduction

Chromosomal data, especially when linked to molecular data in an evolutionary perspective, can be useful to detect plesiomorphic and apomorphic character states, identify different lineages and help to reconstruct evolutionary trends at different taxonomic levels (see e.g., [1,2]). In general, chromosome rearrangements (or macromutations) may either precede or follow molecular differentiation, and they may cause cladogenesis or, conversely, be a result of phylogenetic diversification [3,4]. In either case, they can be treated as discrete markers in evolutionary and phylogenetic studies, highlighting the occurrence of different pathways of karyological diversification [5,6]. Karyotype mutations, such as the acquisition of a different ploidy, inversions or other rearrangements can drive speciation by promoting reproductive isolation (see e.g., [7,8]) and different chromosome states and markers can be useful taxonomic indicators in phylogenetically closely related taxa and in some genome manipulations (see e.g., [9,10,11,12]).

Among vertebrates, amphibians display peculiar genomic and karyological features, including a distinctively large genome size (mostly due to a high heterochromatin content), the occurrence of auto- and allopolyploid lineages, different genetic sex determination systems, and different features (e.g., number, chromosome location) of several chromosome markers (see e.g., [3,13,14]). The Eurasian true salamanders of the family Salamandridae currently include 126 species which are subdivided into the three distinct subfamilies Pleurodelinae, Salamandrininae and Salamandrinae [15]. The latter is composed of five distinct genera: *Mertensiella*, *Chioglossa*, *Lyciasalamandra*, *Salamandra* and the extinct *Megalotriton* (see e.g., [16]). The Lycian salamanders were originally described as *Molge luschani* by [17] and later transferred to the genus *Mertensiella* [18]. During the next 70 years another eight taxa were identified and classified as subspecies of *M. luschani*. However, molecular analyses retrieved *Mertensiella* to be a polyphyletic group [19,20,21] and proposed the creation of the new genus *Lyciasalamandra* [22]. The genus *Lyciasalamandra* is characterized by a debated taxonomy and is currently composed of seven species and 21 subspecies [23,24,25,26,27,28]: *L. atifi* (6 ssp), *L. billae* (5 ssp), *L. fazilae* (2 ssp), *L. flavimembris* (2 ssp), *L. helverseni* (monotypic) and *L. luschani* (3 ssp). Three subspecies of *L*. *billae* (*irfani*, *arikani* and *yehudahi*) had been described initially as full species [29,30], but molecular data suggest considering them as subspecies [25].

Several molecular phylogenetic studies have tried to resolve the phylogenetic relationships within *Lyciasalamandra* [21,25,31,32], however, all of them resulted in a basal polytomy. Veith et al. [24] therefore tested for a scenario of synchronous evolution (and the described polytomy was considered a hard one), which finally they could not reject. As already shown for different taxa at different taxonomic levels, the historical biogeography of Palearctic vertebrates has been greatly influenced by the Quaternary climatic oscillations and related changes of geomorphological features [33,34]. In the case of the Lycian salamanders, molecular studies suggest that the intrageneric diversification of the genus *Lyciasalamandra* was probably triggered by the final emergence of the mid-Aegean trench (10.2–12.3 mya) [25,32]. Similarly, processes of intraspecific diversification (e.g., within *L. luschani*) temporarily correspond to the Messinian Salinity Crisis 5.3 mya [25,32].

In contrast to the growing number of molecular data on *Lyciasalamandra* and the emerging, progressively clearer evolutionary and biogeographic scenario, there are currently no published karyotypes of the genus, leaving their chromosomal features completely unexplored. In this study we performed a comparative cytogenetic analysis on several taxa of *Lyciasalamandra*, providing the first karyological assessment on the genus and highlighting the occurrence and distribution of different conserved and derived chromosomal features. For comparison, and to add unpublished information on their karyotype structure, we also included in our experimental analysis different taxa of the genus *Salamandra*, which is considered the sister taxon to *Lyciasalamandra* [25,32]. Finally, we superimposed our newly generated karyological data on available phylogenetic inferences, comparing alternative topologies retrieved with different datasets, in order to evaluate the possible contribution of chromosome characters on the taxonomy and phylogenetic diversity of the Lycian salamanders. We highlight that the occurrence of a peculiar, taxon-specific pattern of chromosome markers reflects the hypothesized scenario of synchronous evolution in the Lycian salamanders and supports the taxonomic validity of the different studied lineages.

## 2. Materials and Methods

### 2.1. Sampling

We studied five out of the seven described species of *Lyciasalamandra*, including seven different subspecies. For comparative purposes we also included in our experimental analyses three taxa of the genus *Salamandra*, which is considered the sister group to *Lyciasalamandra* [25,32]. A complete list of the samples studied, including sex, origin, number and taxonomic attribution is reported in Table 1. All samples used in this work have already been used in previous molecular and phylogenetic studies [25,27,31,35], where their taxonomic attribution was genetically determined.

### 2.2. Cytogenetic Analysis

All the studied specimens were preliminarily injected with 1 mg/mL colchicine solution (0.1 mL/10 g body weight) for 24 h. After anesthetization in a 0.1% of Tricaine methanesulfonate (MS-222) solution (Sigma-Aldrich), tissue and organ samples (intestine, testis, spleen and kidney) were incubated for 30 min in a 0.7% sodium citrate solution. The organs were fixed for 30 min in Carnoy’s solution (methanol/acetic acid, 3:1). Chromosomes were prepared according to the standard air-drying method following Sidhom et al. [36] and metaphase plates were stained with traditional coloration (5% Giemsa’ solution at pH 7). The determination of karyotypes, relative length (RL) (length of a chromosome/total chromosome length) and centromeric index (CI) (length of the short arm/total length of the chromosome) (see Appendix A) were performed using ten metaphase plates per studied sample and chromosomes were classified following Levan et al. [37]. Chromosomes were then stained with the following banding methods: Chromomycin A_3_ (CMA_3_) + Methyl Green according to the method by Sahar and Latt [38]; Ag-NOR staining following Howell and Black [39] and sequential C-banding following Sumner [40], but using Ba(OH)_2_ at 45 °C and sequentially staining the slides with Giemsa and DAPI according to Mezzasalma et al. [14].

## 3. Results

### 3.1. Chromosome Number and NOR Configuration

All the examined samples of *Lyciasalamandra* and *Salamandra* have a very similar karyotype composed of 2n = 24 biarmed chromosomes, with a prevalence of metacentric pairs (1–5, 7, 9–11) and a lower number of submetacentric pairs (6, 8 and 12) (Figure 1 and Figure 2, Appendix A). An exception is represented by *S. s. salamandra* where the eighth pair is metacentric (Figure 2). In all the taxa with a sample size of n > 1, we found no difference in chromosome number or morphology among different samples. This is also true for *L. l. luschani* and *S. s. gigliolii*, of which specimens from different populations were studied. No differences in chromosome morphology were observed between traditional Giemsa’s coloration (not shown), CMA_3_ staining (Figure 1 and Figure 2) and sequential C-banding (Figure 3 and Figure 4).

CMA_3_ and Ag-NOR staining evidenced in the different studied taxa the occurrence of either two (homomorphic or heteromorphic) or three (two paired and one unpaired) NORs (Figure 1 and Figure 2). We did not detect any difference in the NOR distribution between the two different methods. Two homomorphic NORs were found in *L. a. antalyana* (on the long arm of the eighth pair) and *L*. *helverseni* (in a peritelomeric position on the short arms of the second pair). Two heteromorphic loci were exhibited by *L. l. billae* (on the short arms of one homologous of the chromosome pairs 6 and 12), *L. f. flavimembris* (in an interstitial position on the long arms of one homologous of chromosome pair 2 and in a peritelomeric position on the long arms of one homologous of chromosome pair 12), *L*. *l*. *luschani* (in a peritelomeric position on the long arms of one homologous of chromosomes of pairs 2 and 12), *L. l. basoglui* (in an interstitial position on the short arms of one homologous chromosome pairs 6 and 12), *L. l. finikensis* (in an interstitial and telomeric position on the short arms of one homologous of chromosome pairs 2 and 6, respectively), and *S*. *lanzai* (in a peritelomeric position on the long arms of one homologous of chromosome pair 8 and in an interstitial position on the short arms of one homologous of chromosome pair 10 (Figure 1 and Figure 2). Three NORs were shown by *S*. *s*. *salamandra* (two homomorphic NORs on the short arm of chromosomes of pair 8, while the third locus in an interstitial position on the long arms one homologous of chromosome pair 2). Three NORs were also shown by *S*. *s*. *gigliolii*, two paired NORs in an interstitial position on the long arms of chromosome pair 10 and a third unpaired locus in a peritelomeric position on the long arms of one homologous of chromosome pair 2 (Figure 2).

### 3.2. Heterochromatin Distribution and Composition

All the studied species and subspecies of *Lyciasalamandra* and *Salamandra* showed an overall similar quantity and distribution of heterochromatin. Solid centromeric C-bands were observed on all chromosomes of the studied taxa. Telomeric heterochromatin is also present on most chromosome pairs of all the studied taxa but resulted as less evident than centromeric heterochromatin (Figure 3). In addition to centromeric and telomeric heterochromatin, *L. a*. *antalyana* and *L*. *b*. *billae* also showed interstitial C-bands on most chromosome pairs, namely: pairs 1–6 showed interstitial C-bands both on the long and short arms, while on pairs 7–8 paracentromeric C-bands were only on the long arm; in pairs 9–12 pericentromeric bands were detected only on the long arms (Figure 3). After sequential C-banding + DAPI staining, both centromeric and paracentromeric C-bands resulted also DAPI positive (Figure 4).

## 4. Discussion

### 4.1. Chromosome Number and Morphology

Salamanders are generally characterized by a strong conservation of the chromosome number and morphology, despite an extensive variation of the genome size [41]. It is therefore not surprising that the chromosome number does not differ among the studied taxa of *Lyciasalamandra* and *Salamandra* (2n = 24). In fact, in the whole family Salamandridae, changes in the chromosome number and/or morphology are mostly limited to a few cases of a reduction in the chromosome number from 2n = 24 to 2n = 22, a condition observed in *Notophthalmus viridescens* and different *Taricha* species [42,43]. Similarly, concerning the chromosome morphology, to date the occurrence of telocentric elements is limited to the genus *Tylototriton* while all the other species so far karyotyped possess a chromosome complement composed of all biarmed (mostly meta- and submetacentric) elements [41,42]. Interestingly, in contrast to the general conserved chromosome morphology in true salamanders, the chromosome pair 8 of two different subspecies of the European fire salamander (*S. s. salamandra* and *S. s. gigliolii*) showed a different morphology (metacentric and submetacentric). The chromosome pair 8 results as submetacentric in all the other true salamanders analyzed so far [3,35,44,45] and the metacentric condition here found in *S. s. salamandra* should be considered a derived state, probably resulting from an intrachromosomal rearrangement such as an inversion or a centromere repositioning.

### 4.2. Heterochromatin Diversity and Distribution

Sequential C-banding evidenced a limited diversification of the heterochromatin distribution in the studied taxa. In fact, *L*. *flavimembris*, *L*. *helverseni*, *L*. *l*. *luschani*, *L*. *l*. *basoglui*, *L*. *l*. *finikensis*, *S. s. salamandra*, *S*. *s*. *gigliolii* and *S*. *lanzai* all showed a very similar C-banding pattern with solid heterochromatic blocks localized on centromeric and telomeric regions of all chromosomes. The two remaining species, *Lyciasalamandra antalyana* and *L*. *billae*, showed a higher heterochromatin content with additional interstitial C-bands on all chromosomes. Interstitial C-bands have been described in several newts and salamanders, including the Caucasian salamander *Mertensiella caucasica* [45], and our data further support that interstitial C-bands can be emerging features of urodele chromosomes [46]. Several molecular studies on the origin and amplification of satellite DNA [47,48,49], which is a major component of telomeric and centromeric heterochromatin [50], proposed an evolutionary hypothesis on the heterochromatin variability in Urodela. According to this hypothesis, initial cycles of amplification of satellite DNA arrays take place at centromeric/pericentromeric regions. Then, as a consequence of following structural intrachromosomal rearrangements, satellite sequences may be dislocated away from the original centromeric regions, eventually also occurring on interstitial and/or telomeric regions [46,47,48]. However, some arrays of the original amplified satellite sequence will be generally still evident on centromeric C-bands as remnants of their original position. In *Lyciasalamandra* and *Salamandra*, the heterochromatin pattern observed in the different studied taxa seems to follow the proposed hypothesis. In fact, the centromeric localization of heterochromatic blocks here found in most of the studied taxa would represent an ancestral condition, while the interstitial heterochromatin in *L. antalyana* and *L. billae* probably corresponds to a derived state originated from following intrachromosomal rearrangements. Furthermore, C-banding + DAPI evidenced that both centromeric and paracentromeric C-bands are mainly constituted of AT-rich sequences, which characterize many different satellite families [51,52,53,54].

### 4.3. Variability of NOR Loci

We detected an impressively high variability in the NOR distribution, highlighting a peculiar taxon-specific configuration in all the studied species and subspecies of *Lyciasalamandra* and *Salamandra*. The correspondence between dot-shaped CMA_3_ positive blocks and NORs has been proved in various studies on different taxa [43,55,56,57] and we found no differences in the NOR distribution with Ag-NOR staining and CMA_3_ or between individuals of the same species or subspecies. Surprisingly, most of the studied taxa (*L*. *billae*, *L*. *flavimembris*, all the studied subspecies of *L. luschani* and *S*. *lanzai*) have two heteromorphic NORs, localized on a single homolog of different chromosome pairs, which represent a very unusual condition. In turn, the occurrence of paired homomorphic NOR loci, as in *L*. *antalyana* and *L*. *helverseni*, is the common state in amphibians and more general in vertebrates [55]. In vertebrates, NORs on non-homologous chromosomes have been found so far only in some Perciformes, such as the pair 2 (^NOR+, NOR−^) and 6 (^NOR+, NOR−^) of the damselfish *Chrysiptera rollandi* [58]. Kasiroek et al. [58] hypothesized that the described NOR heteromorphism originated from translocations but did not infer how the observed peculiar NOR phenotype became fixed in the species. In general, two tentative explanations can be advanced to account for the peculiar NOR configuration found in *Lyciasalamandra* and *Salamandra*. The first is related to meiotic mechanisms that could promote a specific chromosome positioning and chromosome-dependent spindle orientation, generating gametes with heteromorphic NORs. Similar mechanisms, even if unusual, have been shown in various taxa, including amphibians [59]. Alternatively, post-meiotic selective pressure may favor the formation of heteromorphic conditions, acting against the complete functional development of homozygotes [9,60,61]. An example of post-meiotic mechanism for chromosome heteromorphism in the family Salamandridae is represented by the genus *Triturus* which shows a heteromorphic chromosome pair 1, a condition caused by developmental arrest in homomorphic condition [48,60]. In *Triturus*, the chromosome pair 1 is lethal in homozygosis and hemizygosis, and normal embryos are produced only in heterozygotic (heteromorphic) condition [48,60]. However, either heteromorphic and homomorphic conditions are present in different studied taxa, and a selection against homozygotes could not be considered a mechanism occurring in all the studied taxa.

In order to test if the observed different NOR configurations reflect evolutionary affinities inferred from DNA sequence data, we superimposed the karyograms of the studied taxa of *Lyciasalamandra* on the topology of recent phylogenetic reconstruction using the most complete molecular dataset [32] (Figure 5A). In addition, we compared these patterns to alternative tree topologies inferred from other molecular and biochemical markers [24,25] which are inherited by the nuclear genome only (Figure 5B,C), since we would expect a closer match between NOR patterns and nuclear trees compared to an organelle tree.

A limited congruence comes from the intraspecific relationships in *L. luschani*, which are fully resolved in the tree of Ehl et al. [32]. The NOR arrangement is heteromorphic in all three subspecies with one NOR-bearing chromosome always shared by two of them (second, sixth or twelfth, respectively), suggesting a mixture of inheritance from a common ancestor and following chromosome rearrangements. However, given the complex pattern of NOR distribution and the lack of intermediate stages, it is difficult to reconstruct possible sequential steps in the phylogenetic diversification of *Lyciasalamandra*. Nevertheless, the occurrence of different NOR patterns represents a useful indicator in the identification of different lineages [62,63]. In this regard, the peculiar taxon-specific NOR configuration of *Lyciasalamandra* and *Salamandra* supports the taxonomic validity of all the studied species and subspecies and highlights that NOR rearrangements are clearly related to events of lineage diversification in the Lycian salamanders. A combination of multiple, independent NOR translocations and other chromosome rearrangements which did not change the overall chromosomal morphology (e.g., paracentric inversions) may have contributed to the highly variable NOR configuration observed in *Lyciasalamandra*. The available cytogenetic information, with hardly any characters shared between two taxa, supports the scenario of a synchronous evolution hypothesized by Veith et al. [24]. These evidences also provide a cytogenetic explanation of the polytomic relationships retrieved in multiple molecular phylogenetic analyses [21,25,31,32]. A more comprehensive sampling including other described species and subspecies may further clarify the debated evolutionary history of *Lyciasalamandra* and the role of chromosome rearrangements in inter- and intraspecific diversification processes of the genus.

## 5. Conclusions

The karyotypes of five species and seven subspecies of *Lyciasalamandra* are here described for the first time and compared with representatives of its sister genus *Salamandra*. All the studied taxa showed a conserved chromosome number (2n = 24), mostly composed of biarmed elements. We detected a limited diversification in heterochromatin content and distribution after sequential C-banding, with a preferential accumulation on centromeric, telomeric and pericentromeric regions. In turn, we found a striking variability in number and location of NORs, with a peculiar taxon-specific configuration of these chromosome markers. Most of the studied taxa (*L*. *billae*, *L*. *flavimembris*, all the studied subspecies of *L. luschani* and *S*. *lanzai*) have two heteromorphic NORs, localized on a single homolog of different chromosome pairs, while paired homomorphic NOR loci were detected in *L*. *antalyana* and *L*. *helverseni*. The peculiar taxon-specific NOR configuration of *Lyciasalamandra* and *Salamandra* supports the taxonomic validity of all the studied species and subspecies and highlights a correlation between NOR rearrangements and events of lineage diversification in the Lycian salamanders. A combination of independent translocations and chromosome inversions may have produced the observed complex NOR configurations, supporting a scenario of synchronous evolution in *Lyciasalamandra*.

## Figures and Tables

**Figure 1 animals-11-01709-f001:**
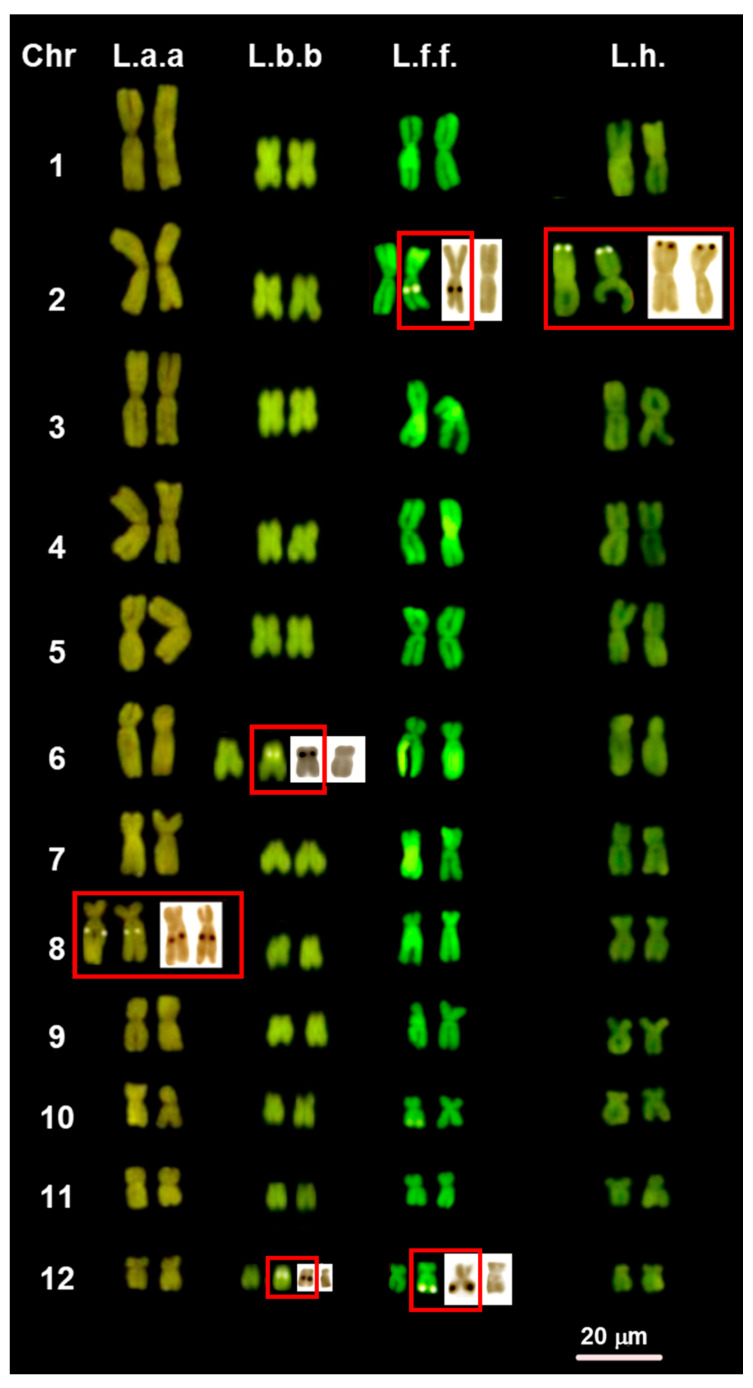
CMA_3_-stained karyotypes of: L.a.a. = *L. a. antalyana;* L.b.b. = *L. b. billae*; L.f.f. = *L. f. flavimembris*; L.h. = *L.*
*helverseni*. NOR-bearing chromosomes stained with CMA_3_ (**left**) and Ag-NOR staining (**right**). The red boxes highlight the occurrence of homomorphic and/or heteromorphic loci.

**Figure 2 animals-11-01709-f002:**
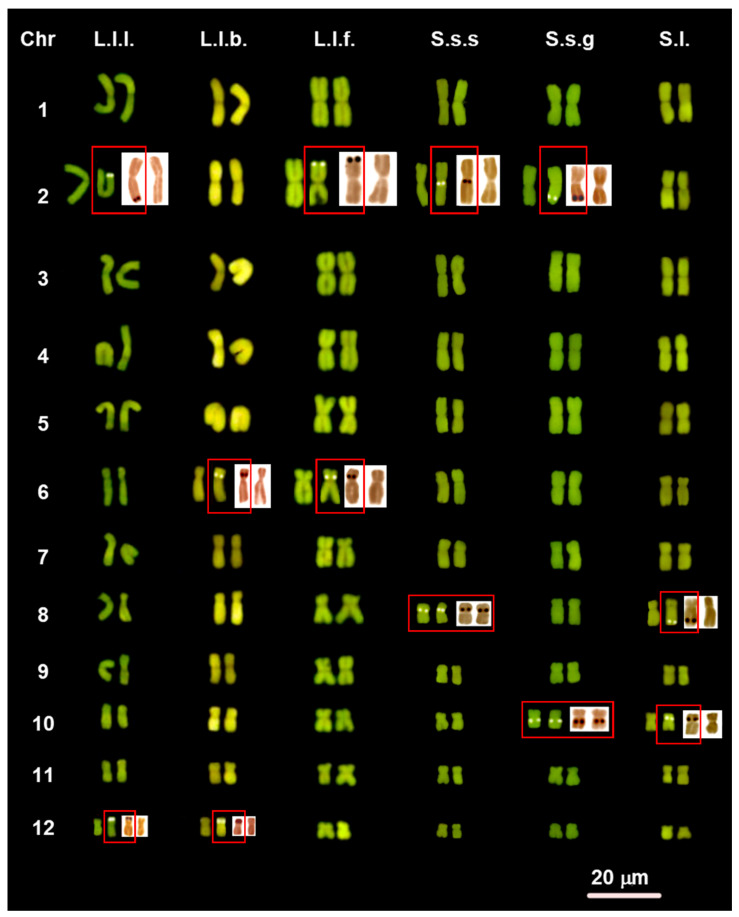
CMA_3_-stained karyotypes of: L.l.l = *L. l. luschani*; L.l.b. = *L. l.*
*basoglui*; L.l.f. = *L. l. finikensis*; S.s.s. = *S. s. salamandra*; S.s.g. = *S. s. gigliolii*; S.l. = *S. lanzai*. NOR-bearing chromosomes stained with CMA_3_ (**left**) and Ag-NOR staining (**right**). The red boxes highlight the occurrence of homomorphic and/or heteromorphic loci.

**Figure 3 animals-11-01709-f003:**
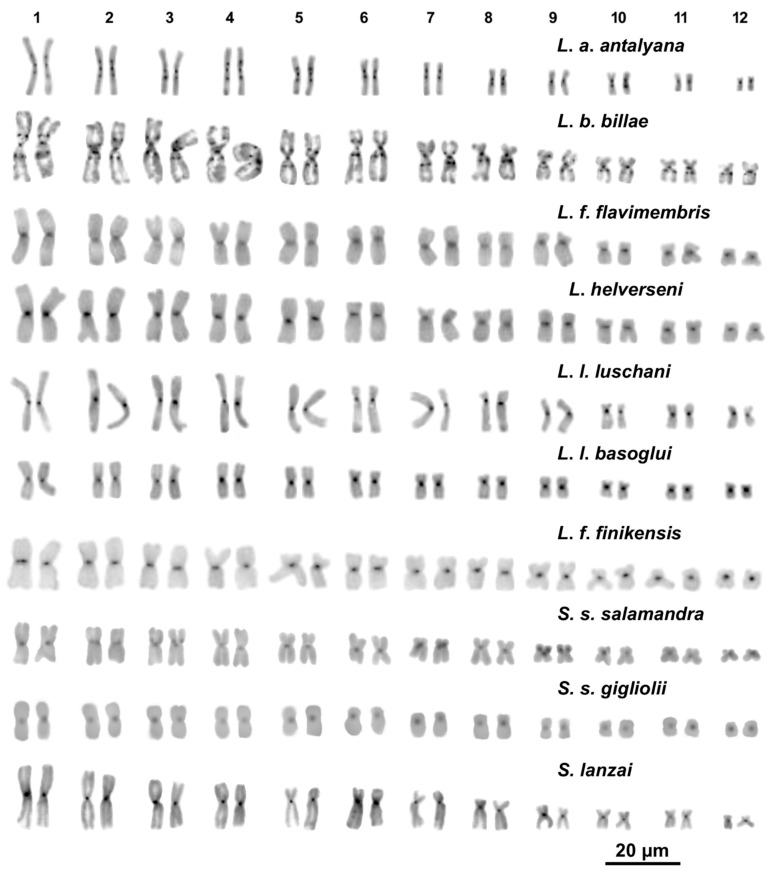
C-banding + Giemsa stained karyotypes of the examined taxa of *Lyciasalamandra* and *Salamandra*.

**Figure 4 animals-11-01709-f004:**
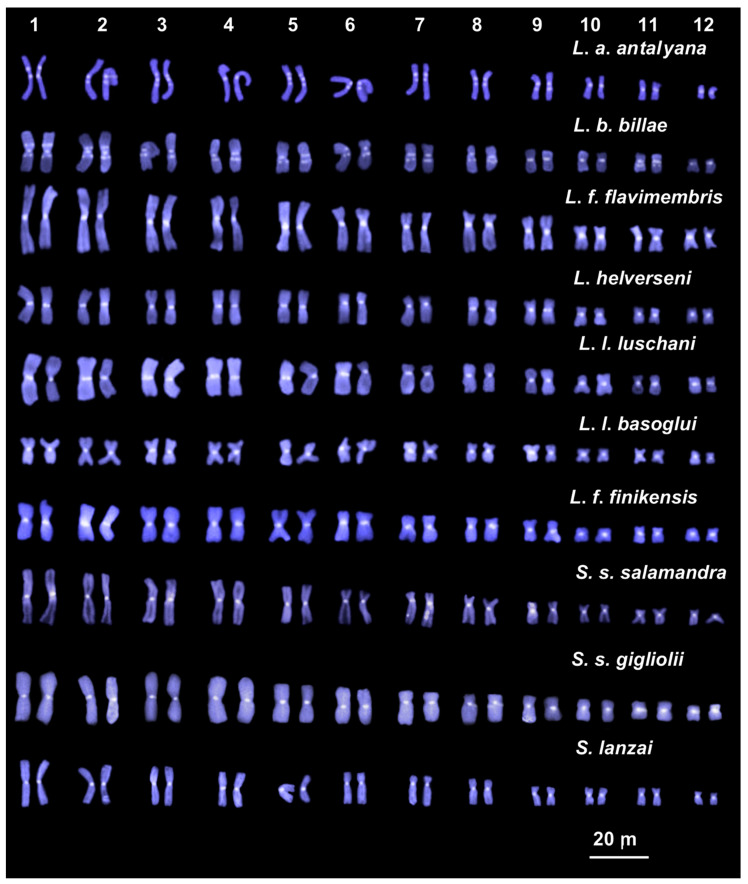
C-banding + DAPI-stained karyotypes of the examined taxa of *Lyciasalamandra* and *Salamandra*.

**Figure 5 animals-11-01709-f005:**
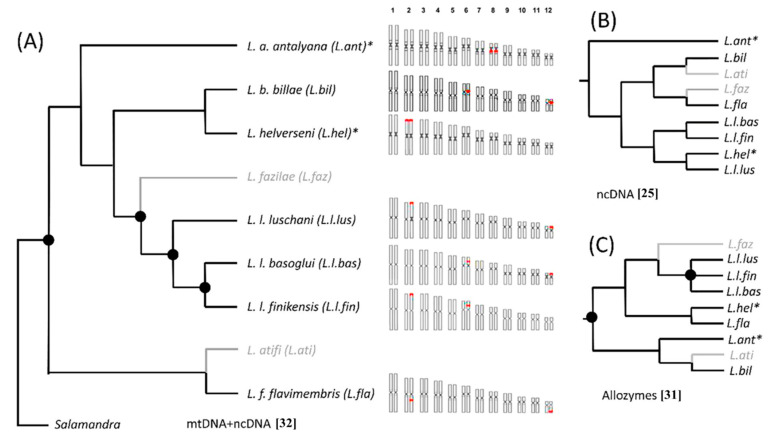
(**A**) Karyograms of the studied taxa of *Lyciasalamandra* superimposed on the phylogenetic relationships redrawn from Ehl et al. [32]; (**B**) alternative topology based on ncDNA [25]; (**C**) alternative topology based on allozymes [31]. Taxa not included in the cytogenetic analysis are given in grey; filled dots indicate nodes with high posterior support values in the original paper; the asterisks indicate taxa with paired homomorphic NOR loci.

**Table 1 animals-11-01709-t001:** Taxonomic attribution, sex and origin of the samples studied.

Genus	Species/Subspecies	Sampling Locality	Number	Sex
*Lyciasalamandra*				
*L.*	*antalyana antalyana*	Hurma (Turkey)	1	♂
*L.*	*billae billae*	Kale Tepe (Turkey)	2	♂
*L.*	*flavimembris flavimembris*	Marmaris (Turkey)	1	♂
*L.*	*helverseni*	Pigadia (Greece)	1	♂
*L.*	*luschani luschani*	Letoon (Turkey)	2	♂
*L*	*luschani luschani*	Dodurga (Turkey)	1	♂
*L.*	*luschani basoglui*	Nadarla (Turkey)	2	♂
*L.*	*luschani finikensis*	Finike (Turkey)	2	♂
*Salamandra*				
*S*.	*salamandra salamandra*	Borgosesia (Italy)	4	♂
*S*.	*salamandra gigliolii*	Serino (Italy)	4	♂
*S*.	*salamandra gigliolii*	Amalfi (Italy)	2	♂
*S*.	*salamandra gigliolii*	Serre (Italy)	3	♂
*S*.	*lanzai*	Germanasca (Italy)	2	♂

## Data Availability

The data presented in this study are available in the manuscript and in Appendix A.

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
