# Peer review of "Karyological Diversification in the Genus Lyciasalamandra (Urodela: Salamandridae)"

_animals, 2021, doi:10.3390/ani11061709_

Round 1
Reviewer 1 Report
The paper from Mezzasalma et al. is interesting and provides new cytological information on Lyciasalamandra that can be useful for understanding the phylogeny of this salamander taxon However it can not be accepted in its present form but several changes are needed.
INTRODUCTION lines 43-47 it could be better to reduce the self-cytotion, but please quote articles of more general interest.
RESULTS: the CMA3 and Ag-NOR reported pictures are very difficult to examine; please add specific figures of CMA3 and of Ag-NOR karyotypes of better quality.
The interstitial heterochromatin bands are not very clear both in C-banded and DAPI stained chromosomes.
DISCUSSION: Lines 235-242 The comparison wth the heteromorphism of the chromosome number 1 in Triturus is useful only to mention an example of post-meiotic mechanism driving chromosome heteromorphism, but it is not completely appropriate because the Triturus heteromorphism does not regard the NOR localization, but length of long arm and level of heterochromatin and is more relevant.
Moreover heteromorphism in NOR localization and number were found in various animal species and generally do not cause great damages.
Lines 265-267 Differences in number and heteromorphism of NORs can not be determined by combination of multiple independent translocations and chromosome inversion because no one species and subsèpecies of Lyciasalamndra show differences in chromosome morphology.
In conclusion I reccomend to accept the paper fron Mezzasalma after the acceptance of the suggeste modification.
Author Response
Reviewer 1
Rev 1: “INTRODUCTION: lines 43-47 it could be better to reduce the self-cytotion, but please quote articles of more general interest”.
Authors’ reply: as suggested, we reduced the self-citations at lines 43-47 and added new references to papers of more general interest (see citations [5,6])
Rev 1:” RESULTS: the CMA3 and Ag-NOR reported pictures are very difficult to examine; please add specific figures of CMA3 and of Ag-NOR karyotypes of better quality.
Authors’ reply: We rearranged, better contrasted and split the image in two different figures (Fig. 1 and 2) so that chromosomes pairs and NOR loci are presented at higher magnification. On the other hand, we prefer to keep together NOR-bearing chromosome pairs stained with both CMA3 and Ag-NOR to better show the co-localization of NOR signals with different methods. In fact, we also tried some alternatives (e.g. splitting CMA3 and Ag-NOR stained karyotypes), but in our opinion it would be more difficult for the reader to follow the NOR co-localization (here highlighted also with the red boxes).
Rev 1: “The interstitial heterochromatin bands are not very clear both in C-banded and DAPI stained chromosomes”.
Authors’ reply: We better contrasted the figures with C-banding + Giemsa and C-banding+DAPI, to optimize the visualization of heterochromatic bands.
Rev 1:” DISCUSSION: Lines 235-242 The comparison with the heteromorphism of the chromosome number 1 in Triturus is useful only to mention an example of post-meiotic mechanism driving chromosome heteromorphism, but it is not completely appropriate because the Triturus heteromorphism does not regard the NOR localization, but length of long arm and level of heterochromatin and is more relevant. Moreover, heteromorphism in NOR localization and number were found in various animal species and generally do not cause great damages.”.
Authors’ reply: We agree, and now we better specify in the Discussion that the comparison with the heteromorphism in Triturus is just to refer to a post-meiotic mechanism for chromosome heteromorphism in the family Salamandridae. We also underline that “.
Rev 1: “Lines 265-267 Differences in number and heteromorphism of NORs can not be determined by combination of multiple independent translocations and chromosome inversion because no one species and subspecies of Lyciasalamndra show differences in chromosome morphology”.
Authors’ reply: We modified the sentence in: “A combination of multiple, independent, NOR translocations and other chromosome rearrangements which did not change the overall chromosomal morphology (e.g. paracentric inversions) may have contributed to the highly variable NOR configuration observed in Lyciasalamandra”.

Reviewer 2 Report
The MS is interesting and well prepared but ..... I did not see the Figures into the text. I marked some somments in the MS file.

Author Response
Reviewer 2
No general comments to address and we followed the specific suggestions annotated by Rev 2 in the pdf file as reported as follows.
Line 51: We added the suggested reference.
Line 124: Sorry, we did not understand very well to what is Ref 2 referring at line 124: “Please preseny any methods of comparison of species”. However, in case the comparison refers to chromosome data we specify in the methods the calculation performed for determination of karyotypes, relative length (RL) and centromeric index (CI) (2.2. Cytogenetic analysis, lines 119-122). In the case the comparison refers to the taxonomic attribution, we stated (in 2.1. Sampling, lines 107-109) that our samples were used in previous molecular and phylogenetic analyses where their taxonomic attribution was genetically determined.
Line 275: We corrected the wrong number.
Line 276: We changed the sentence to passive form.

Round 2
Reviewer 2 Report
The MS was corrected due to Reviewers suggestions. So, I recommend accepting MS for publication as it stands.